# Development and Effectiveness of a Pattern Management Educational Program Using Continuous Glucose Monitoring for Type 2 Diabetic Patients in Korea: A Quasi-Experimental Study

**DOI:** 10.3390/healthcare12141381

**Published:** 2024-07-10

**Authors:** Seung-Yeon Kong, Mi-Kyoung Cho

**Affiliations:** 1Referral Center, Chungbuk National University Hospital, Cheongju 28644, Republic of Korea; rrhd2@cbnuh.or.kr; 2Department of Nursing Science, Research Institute of Nursing Science, Chungbuk National University, Cheongju 28644, Republic of Korea

**Keywords:** type 2 diabetes, self-care, continuous glucose monitoring, pattern management, glycemic control

## Abstract

Background: The prevalence of diabetes has increased worldwide. Therefore, interest in improving glycemic control for diabetes has grown, and continuous glucose monitoring (CGM) has recently received attention as an effective glycemic control method. This study developed and evaluated the effectiveness of an education program for pattern management using CGM based on Whittemore and Roy’s middle-range theory of adapting to diabetes mellitus. Methods: A quasi-experimental study was conducted on 50 adult patients with type 2 diabetes who visited the outpatient clinic of a university hospital. The experimental group was treated with a pattern management program using CGM for 12 weeks and six personalized education sessions were provided to the patients through face-to-face education and phone monitoring. Results: The frequency of diabetes-related symptoms in the experimental group decreased, and social support (*t* = 2.95, *p* = 0.005), perceived benefits (*t* = 3.72, *p* < 0.001) and self-care significantly increased (*t* = 6.09, *p* < 0.001). Additionally, the program was found to be effective in improving HbA1c (*t* = −3.83, *p* < 0.001), FBS (*t* = −2.14, *p* = 0.038), and HDL-C (*t* = 2.39, *p* = 0.021). Conclusion: The educational program developed through this study can be implemented as a self-management approach for individuals with type 2 diabetes using CGM, aimed at enhancing glycemic control and preventing complications.

## 1. Introduction

According to the International Diabetes Federation, the worldwide adult population (aged 20–79 years) with diabetes mellitus was 537 million in 2021, and the condition’s prevalence is continuing to rise, exceeding previously predicted growth rates by nearly 12 years. By 2045, the number of individuals with diabetes is expected to reach 783 million [1]. The prevalence of diabetes in Korea is growing at a faster rate than expected, with approximately 6 million adults (aged ≥30) having diabetes as of 2022, a number that the Korean Diabetes Association originally predicted would be reached by 2050 [2]. 

Diabetes is a common condition in which poor glycemic control leads to problems in the macro and microvascular systems, resulting in various serious complications [3]. In Korea, 84.7% of diabetic patients aged ≥30 developed chronic complications, with the prevalence of coronary artery disease ranging from 14.1% to 16.8% and cerebrovascular disease ranging from 8.8% to 11.4% [4]. To prevent these complications, proper glycemic control is required; according to a report by the Korean Diabetes Association in 2022, only 24.5% of diabetics in Korea have their blood glucose controlled at an appropriate level, and only 9.9% have their blood glucose, blood pressure, and cholesterol maintained at target levels. This finding emphasizes the need for active and integrated interventions for glycemic control in diabetes.

Self-monitoring of blood glucose (SMBG) remains one of the most effective approaches for a long-term glycemic control in people with diabetes [5]. However, these methods are painful and cumbersome, requiring patients to puncture their fingers multiple times a day [6]. Additionally, limitations have been reported regarding understanding changes in specific blood glucose patterns, such as postprandial blood glucose elevations or asymptomatic hypoglycemia [7]. 

Recently, continuous glucose monitoring (CGM), which compensates for SMBG limitations by inserting a sensor subcutaneously in diabetic patients to measure the glucose level of the interstitial fluid every 1–5 min to provide blood glucose levels and changes in real time [8,9], has been effectively applied to patients with diabetes in various situations [10], and its use has expanded over the past few years with the increasing accuracy of the sensors and improved convenience of use [11]. Previous studies reported improved self-care behaviors, improved glycemic control, and reduced HbA1c levels in patients with CGM [12,13,14,15]. However, the use of CGM without specialized training was reportedly ineffective in improving glycemic control or reducing severe hypoglycemia [16], suggesting that healthcare provider interventions, along with specialized training, are critical to ensure the proper use of CGM by patients [17].

Menti et al. [18] found greater improvement in self-care when theory-based education was provided to type 2 diabetics. Therefore, this study developed a program based on the adaptation to the diabetes mellitus middle-range theory developed by Whittemore and Roy [19] for patients with diabetes. This theory explains how patients with diabetes adjust to living with diabetes and suggests that improved self-care will help them adjust to daily life and improve their glycemic control, allowing them to live to their full health potential. Given that no previous studies applied and verified this theory among patients with diabetes, this study aimed to develop a pattern management educational program using CGM to improve self-care, based on this theory, in patients with diabetes and to verify its effectiveness.

## 2. Materials and Methods

### 2.1. Research Design

This quasi-experimental study aimed to implement a pattern management educational program using CGM (FreeStyle Libre; Abbott Diabetes Care, Witney, UK) and to verify its effectiveness in type 2 diabetic patients. 

### 2.2. Participants

This study included patients diagnosed with type 2 DM attending the outpatient department of endocrinology at a university hospital in C city. The inclusion criteria were as follows: aged ≥18 years, HbA1c level of ≥6.5%, receiving insulin injections for diabetes treatment, understanding the purpose and method of the study, providing written consent to participate, being able to read and understand the questionnaire content, and completing both pre- and post-questionnaires. The exclusion criteria were as follows: patients who used insulin pumps, were hospitalized with diabetes complications, were unable to self-administer insulin, or had difficulty understanding or manually completing the questionnaires. The sample size was determined utilizing the G*Power 3.1 software [20] with an effect size (d) of 0.81, significance level (α) of 0.05, power (β) of 0.80, and two-tailed independent *t*-test based on previous studies [21]. Consequently, a total of 50 participants (25 in each group) were required, and 58 participants were intended to be recruited, considering a 15% dropout rate.

Type 2 diabetic patients who were prescribed CGM and were willing to participate, after being introduced to the study program by their professors, were assigned to the experimental group, whereas those who did not want to use CGM were assigned to the control group. In this study, the device’s accuracy was verified by comparing the glucose levels measured by the CGM (FreeStyle Libre) with those obtained from finger-prick blood samples during the first week after sensor insertion. After we excluded 2 patients who did not meet the inclusion criteria and 4 patients who refused to participate after being informed about the study’s duration and methods, 52 patients were included in the study, with 26 in the experimental group and 26 in the control group. During this study, one patient in the experimental group moved out of the country and one patient in the control group passed away from another condition, leaving 25 patients in each of the experimental and control groups; therefore, a total of 50 patients were included in the final analysis (Figure 1).

### 2.3. Data Collection and Procedure

Data collection was conducted from 7 June 2022 to 26 September 2023, with the approval of the Institutional Review Board of the investigator’s institution (NO: 2022-05-027-001), and informed consent was obtained from the professor of endocrinology at the hospital after we explained the study’s purpose and process. The pattern management educational program using CGM was initially evaluated by a multidisciplinary team of experts, including an endocrinology professor, a nursing professor, and experienced diabetes care nurses. They assessed the program’s structure and content using established criteria, with all components achieving validation scores above 0.80, indicating strong content validity and reliability. A pilot study, distinct from our main study sample, was conducted for three months with three patients from the outpatient clinic at the same hospital. After the pilot intervention, significant improvements were noted in HbA1c levels, fasting blood sugar, and self-care scores among all subjects, affirming the potential of this educational program to effectively enhance glycemic control in type 2 diabetes. After the pilot study, the patients who participated in the pilot study were individually interviewed by an investigator. This preliminary phase served as an initial test to refine our intervention strategies based on direct feedback and measurable improvements in key diabetes metrics such as HbA1c and fasting blood sugar levels, as well as self-care behaviors. The insights gained from this pilot study informed further refinements to the program, enhancing its educational content and delivery methods. 

Before the implementation of the program, a pre-test of sociodemographic characteristics, disease-related characteristics, previous health promotion behavior, illness perception, diabetes symptoms, diabetes social support, diabetes health beliefs, and self-care was conducted using a self-report questionnaire for all patients in the experimental and control groups, and their physiological indices (BMI, HbA1c, FBS, cholesterol, LDL-C, HDL-C, triglyceride, SBP, and DBP) were examined.

A post-test on diabetes symptoms, social support, health beliefs, and self-care was conducted during outpatient visits for regular follow-up within 2 weeks after the end of the 12-week program, and their physiological indices were examined. 

### 2.4. Program Development

This study’s pattern management educational program using CGM was developed to provide one-on-one personalized education for effective and sustained self-care and improvement of physiological indices in type 2 diabetic patients, aiming to help patients identify and correct factors contributing to poor glycemic control and adjust their insulin doses professionally to achieve target blood glucose levels. This program included one-on-one education by the researchers who analyzed the blood glucose results transmitted through the CGM sensor and the investigator was trained in theory and practice to provide professional education. Additionally, the participants’ educational needs were identified through professional interviews, and the program was developed based on theory to increase the level of evidence for the intervention (Figure 2) [22]. The composition of the program, according to the theoretical framework (Figure 3), is presented in Table 1, and education was provided by classifying the pattern according to the type of insulin to provide an accurate diagnosis and specific intervention according to the blood glucose pattern.

### 2.5. Intervention

Regarding the first week, after the preliminary survey, the experimental group received education on understanding and using CGM, and the CGM sensor was placed on the posterior aspect of the upper arm, as this is an area with high subcutaneous fat. During the next 14 days, the participants engaged in daily activities with the sensor inserted, during which time they were monitored every 3 days to ensure that they were using the CGM correctly with a full understanding of the information provided in the initial training. After the 14 days, during week 3, the sensor was removed, and the AGP report with 14 days of recorded blood glucose was printed to understand the blood glucose pattern, and personalized education on insulin, diet, and exercise was provided. After the face-to-face education, the patients were monitored via mobile phone every two weeks (weeks 5, 7, 9, and 11) to examine their glycemic control status and the extent of improvement in any problematic factors, provide feedback, answer questions, provide education on newly identified problems, and encourage adherence (Table 1). The control group adhered to their existing treatment regimen.

### 2.6. Outcome Assessment

#### 2.6.1. Diabetes Symptoms

Diabetes symptoms include polyuria, polydipsia, polyphagia, weight loss, and several nonspecific symptoms, including headache and fatigue [11], which were measured using the Diabetes Symptom Self-Care Inventory (DSSCI) developed by García [23] and adapted and validated in Korean by Hong and Yoo [24]. It consists of 20 items in five domains rated on a 5-point Likert scale. Higher scores indicate a higher frequency of symptoms in each cluster. The reliability of the scale, Cronbach’s α, was 0.81 at the time of development by García [23], 0.94 in the study by Hong and Yoo [24], and 0.89 in this study.

#### 2.6.2. Diabetes Social Support

Diabetes social support refers to diabetes intensive care therapy that affects patients with diabetes and their caregivers [25], which was measured by the diabetes-specific social support scales from the Diabetes Care Questionnaire (Diabetes Care Profile, DCP) developed by Fitzgerald et al. [26] and adapted and validated in Korean by Byun [27]. It consists of six items on a 5-point Likert scale. Higher scores indicate a higher level of diabetes social support. The reliability of the scale, Cronbach’s α, was 0.73 at the time of development by Fitzgerald et al. [26], 0.87 in the study by Byun [27], and 0.73 in this study.

#### 2.6.3. Diabetes Health Beliefs

Health beliefs refer to individuals’ subjective beliefs that underlie the behaviors they undertake to prevent disease [28], which this study measured using the diabetes health belief scales developed by Bak [29]. It consists of 18 items in three domains rated on a 5-point Likert scale. Higher scores indicate higher levels of perceived susceptibility, severity, barriers, and benefits of diabetes. At the time of scale development by Bak [29], the reliability of the scale, Cronbach’s α, was 0.76 for susceptibility and severity, 0.65 for barrier, and 0.69 for benefit. The Cronbach’s α for this study was 0.78 for susceptibility and severity, 0.68 for barrier, and 0.67 for benefit. 

#### 2.6.4. Self-Care

Self-care refers to goal-directed activities in which individuals perform continuously and voluntarily to maintain their life, health, and well-being [30]. Self-care was measured by the self-care of diabetes inventory (SCODI) developed by Ausili et al. [31], revised in Korean, and validated for reliability and validity by this study’s investigator [32]. It consists of 40 items in four domains rated on a 5-point Likert scale. Higher scores indicate a higher level of self-care in each domain. At the time of scale development by Ausili et al. [31], the reliability of the scale, Cronbach’s α, was 0.81 for maintenance, 0.84 for monitoring, 0.86 for management, and 0.89 for confidence. The Cronbach’s α was 0.77 for maintenance, 0.69 for monitoring, 0.81 for management, and 0.92 for confidence in the study by Kong and Cho [32], and 0.67 for maintenance, 0.74 for monitoring, 0.79 for management, and 0.94 for confidence in this study.

#### 2.6.5. Physiological Indices

Physiological indices refer to measured variables that are related to human biological phenomena [33], This study identified body mass index (BMI), glycosylated hemoglobin (HbA1c), fasting blood sugar (FBS), cholesterol, low-density lipoprotein cholesterol (LDL-C), high-density lipoprotein cholesterol (HDL-C), triglycerides, systolic blood pressure (SBP), and diastolic blood pressure (DBP) as diabetes-related physiological indices, and the results were collected from medical records. Blood pressure was measured using an Omron automatic electronic sphygmomanometer HEM7130 (Omron Healthcare Manufacturing Co., Ltd., Thu Dau Mot, Vietnam). BMI was measured using a BSM330 (Biospace, Seoul, Republic of Korea), HbA1c was measured using an HbA1c analyzer (HbA1c HA-8180, Arkray, Inc., Kyoto, Japan), and FBS and lipid profiles were measured using an automatic chemical analyzer (TBA-FX8 Module 1; Toshiba, Inc., Tokyo, Japan).

### 2.7. Statistical Analysis

The data were analyzed using SPSS (IBM SPSS statistics version 26; IBM Inc., Chicago, IL, USA). The demographic and disease-related characteristics of the participants were analyzed using descriptive statistics; the Shapiro–Wilk test was conducted to determine the normality of the experimental and control groups; the Chi-square test, the independent *t*-test, and Fisher’s exact test were conducted to examine the pre-homogeneity of the two groups; and an independent *t*-test was conducted to examine differences between the two groups after the intervention. The statistical significance was set at *p* < 0.05.

## 3. Results

### 3.1. Participant Characteristics at Baseline and Homogeneity Test

The study included a total of 50 participants, 25 in each group. Regarding sex, there were 14 (56.0%) females in the experimental group and 15 (60.0%) in the control group, with a mean age of 53.96 ± 9.77 years in the experimental group and 54.92 ± 11.73 years in the control group. The duration of diabetes was 12.39 ± 8.30 years in the experimental group and 10.31 ± 8.12 years in the control group, and the most common treatment of diabetes was a combination of oral hypoglycemic agents and insulin, which was used by 16 participants (64.0%) in the experimental group and 15 (60.0%) in the control group. 

The HbA1c was 10.76% ± 2.78% in the experimental group, which exceeded that of the control group (9.60% ± 1.92%); however, no statistically significant difference was observed. The diabetes self-care score was 63.95 ± 11.30 in the control group, which exceeded that of the experimental group (56.36 ± 16.48); however, there was no significant difference. 

In the above analysis, the homogeneity test for sociodemographic characteristics, disease-related characteristics, main variables, and physiological indices of the participants indicated no statistically significant differences between the two groups, indicating that the experimental and control groups were homogeneous (Table 2).

### 3.2. Outcome Assessment

As a result of measuring the frequency of diabetes symptoms after the intervention, a significant decrease in the frequency of psychological–cognitive symptoms (*p* < 0.001), thirst–fatigue symptoms (*p* = 0.049), neurological symptoms (*p* = 0.015), and gastrointestinal comfort symptoms (*p* < 0.001) were observed in the experimental group compared with the control group; however, no change in the sexual symptoms was observed in either group. 

The experimental group exhibited a significantly higher diabetes social support score compared to the control group (*t* = 2.95, *p* = 0.005). 

Regarding diabetes health beliefs, the perceived benefits were significantly higher in the experimental group than in the control group (*p* < 0.001); however, no differences were observed in perceived susceptibility and severity or perceived barriers. 

Additionally, a significantly higher self-care score was observed in the experimental group than in the control group (*t* = 6.09, *p* < 0.001). Regarding domains, a significant increase was observed in maintenance (*t* = 5.29, *p* < 0.001), monitoring (*t* = 5.04, *p* < 0.001), management (*t* = 4.42, *p* < 0.001), and confidence (*t* = 2.99, *p* = 0.004). 

As a result of testing the difference in changes in diabetes-related physiological indices, HbA1c (*t* = −3.83, *p* < 0.001) and FBS (*t* = −2.14, *p* = 0.038) were significantly reduced and HDL-C was significantly increased (*t* = 2.39, *p* = 0.021) in the experimental group compared with the control group after the program. No significant differences were observed in cholesterol, LDL-C, triglyceride, or blood pressure between the groups (Table 3). 

## 4. Discussion

This study developed and validated the effectiveness of a pattern management educational program using CGM to improve self-care and physiological indices in type 2 diabetics receiving insulin therapy.

The experimental group, which participated in the pattern management educational program using CGM, showed a decrease in psychological–cognitive, thirst–fatigue, neurological, and gastrointestinal comfort symptoms compared with the control group, which adhered to the usual diabetes treatment regimen. Given that diabetes symptoms are related to glycemic control, these results may have been attributed to the participants’ improvement in physiological indices, such as FBS, HbA1c, and HDL-C, after the intervention. Psychological–cognitive symptoms, which were the most common symptoms experienced by the participants before participating in the program, decreased the most after the intervention [11]. When diabetic patients become hypoglycemic, they may experience psychological symptoms such as anxiety, palpitations, sweating, irritability, anger, and sleep disturbances [11]. The most significant reduction in these symptoms appears to have been due to the prevention of hypoglycemia during the 12-week intervention period. Given that no patients at the hospital had hypoglycemia during the study, this intervention appears to be helpful for patients on insulin therapy who fear hypoglycemia. 

Furthermore, social support increased in the experimental group after this study’s intervention program. In this study, when providing education via phone to patients aged ≥65 years who needed to adjust their insulin dose, they sometimes put the phone on speaker to allow family members to also receive the education and help with dosing to ensure clear communication. Azami et al. [34] found that social support increased in type 2 diabetics after a nurse-led diabetes self-care education program that combined face-to-face education and over-the-phone monitoring for 12 weeks, similar to this study. A meta-analysis by Song et al. [35] found that social support was significantly associated with self-care, with the largest effect on glucose monitoring among the types of self-care. Several previous studies confirmed social support as a factor influencing self-care in type 2 DM [36,37,38]. Therefore, social support for patients with diabetes should be considered when providing interventions to improve self-care. 

In this study, scores in the benefit domain of health beliefs improved in the experimental group after the intervention. Shabibi et al. [39] found that benefits improved after a four-week educational program to improve self-care in people with type 2 diabetes, increasing their motivation to perform self-care. Swaleh and Yu [40] described positive health beliefs as an important factor influencing the improvement of glycemic control and self-care in patients. This program may have enhanced patients’ knowledge about diabetes, improved their glycemic control, and increased their awareness of diabetes as a self-controllable disease through repeated education to analyze and correct problems in self-care based on the glycemic control of individuals over a 12-week period. However, no significant differences were observed in susceptibility, severity, or barriers, which corresponds with the findings of Lee [21]. Lee [21] suggested that it was difficult to change the health beliefs of patients with diabetes in a 12-week intervention and that interventions lasting longer than 12 weeks should be considered. 

After the intervention, a significant improvement in self-care scores of the experimental group was observed. In particular, the scores in the self-care monitoring and maintenance domains improved significantly compared to that in the preliminary survey, which measured the ability of patients with diabetes to recognize symptoms, monitor their health status, and take appropriate action to address perceived hyperglycemia, hypoglycemia, or other abnormalities, making these two domains the most critical aspects of self-care for diabetes. This program seemingly improved patients’ ability to identify and resolve factors that are problematic for glycemic control by analyzing their blood glucose patterns through CGM, identifying and correcting problem factors, and providing repeated education to examine improvement levels. Therefore, this program can be considered an effective intervention for improving self-care among patients with type 2 DM. 

Finally, among the physiological indices of the experimental group participating in the pattern-management educational program using CGM in this study, HbA1c and FBS decreased, and HDL-C increased. Given that the management of HbA1c, FBS, and lipid profiles is essential for slowing the progression of the disease and preventing complications in type 2 diabetes [3], this study is meaningful as an intervention to improve diabetes-related physiological indices. Furthermore, when the Korean Diabetes Association [3] published its revised diabetes guidelines, it added an indicator of CGM devices to the glycemic control target and recommended the use of real-time CGM for people with type 2 diabetes treated with multiple insulin injections, as well as emphasizing the need for thorough diabetes management and training, including insulin dosing and proper device usage education for patients. However, the support of the National Health Insurance Service for CGM in Korea is currently only provided to people with type 1 diabetes [41], preventing patients with type 2 diabetes from benefiting from it and limiting their use owing to cost. In response, the National Health Insurance Service launched a study in April 2023 to implement effective policies to expand the coverage of CGM devices for type 2 diabetes [42]. Therefore, this study may serve as a basis for policies to expand the coverage of CGM devices in type 2 DM. 

This study had several limitations. First, this was a quasi-experimental study with a convenience sample, which may have limited the generalizability of the findings to all patients with type 2 diabetes. Second, the study’s results must be interpreted with caution due to potential biases stemming from the participant selection. The research involved type 2 diabetes patients from a university hospital’s outpatient clinic, who regularly received diabetes management education. This consistent educational support may have made them more amenable to the interventions, potentially influencing their responsiveness in ways that might not be representative of the broader diabetic population. Third, this study identified changes in physiological indices as post-intervention outcome variables but did not improve the metrics of the CGM devices. Therefore, it is limited in explaining changes in patients’ blood glucose variability, target blood glucose ratio, hypoglycemia, and hyperglycemia rates. Future studies should identify the metrics of CGM devices as outcome variables. Third, given that this study was conducted on patients visiting a single university hospital, its generalizability is limited; therefore, a multicenter study is recommended.

## 5. Conclusions

The prevalence of DM is increasing all over the world. In Korea, it is increasing at a faster rate than predicted by the Korean Diabetes Association, emphasizing the need for national management solutions. Consequently, there is a growing interest in ways to improve blood glucose management in diabetes, and CGM has emerged as the most important method of glucose monitoring for blood glucose management. Therefore, this study developed a pattern management educational program using CGM based on the adaptation to diabetes mellitus middle-range theory developed by Whittemore and Roy [19] and verified its effectiveness. After the intervention, diabetes symptoms decreased in the experimental group, whereas social support and perceived benefits increased significantly. In addition, self-care and diabetes-related physiological indices such as HbA1c, FBS, and HDL-C improved in the experimental group. Therefore, the program developed in this study can be used as a method of personalized self-care education to improve glycemic control and prevent complications in type 2 diabetes using CGM and can also be used as a basis for policies to expand the insurance coverage of CGM devices for patients with type 2 diabetes.

## Figures and Tables

**Figure 1 healthcare-12-01381-f001:**
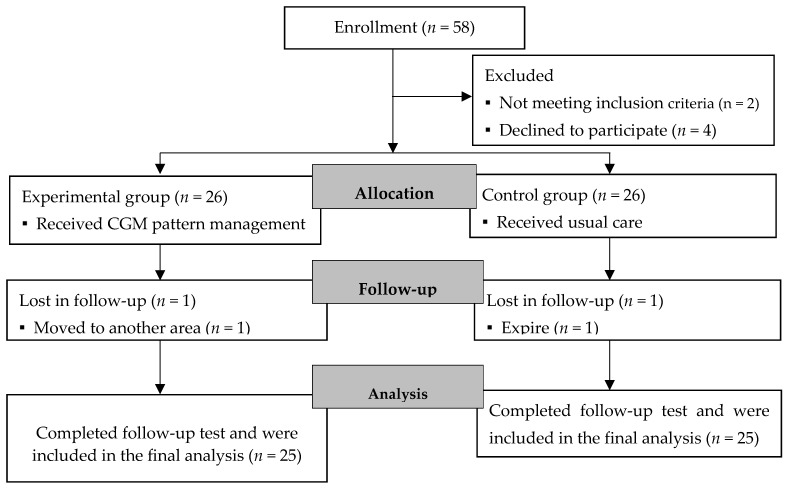
Flow diagram of participant. Note. CGM: continuous glucose monitoring.

**Figure 2 healthcare-12-01381-f002:**
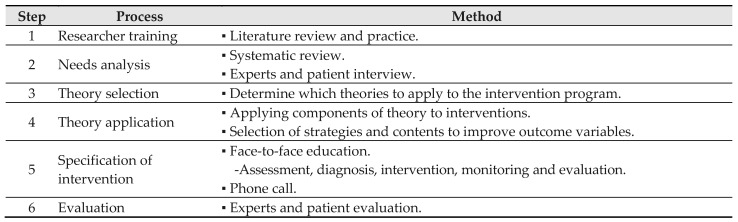
Development process of the pattern management educational program using continuous glucose monitoring.

**Figure 3 healthcare-12-01381-f003:**
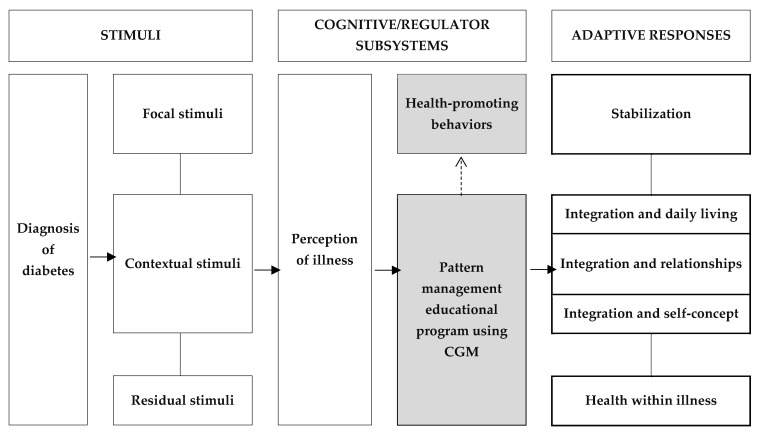
Conceptual framework of this study. Note. CGM = continuous glucose monitoring.

**Table 1 healthcare-12-01381-t001:** Overview of educational program based on adapting to diabetes mellitus theory.

Week	Time	Content	Method	Theoretical Components
1st	30 min	▪ Sociodemographic characteristics survey. ▪ Disease-related characteristics survey. ▪ Previous health-promoting behavior survey. ▪ Illness perception survey.	Questionnaire	Stimuli and perception of illness
		▪ CGM principle. ▪ Necessity and advantages of CGM. ▪ Sensor insertion and precautions. ▪ If self-care of blood glucose is necessary. ▪ Self-care records for checking glucose patterns.	Leaflet	Health-promoting behaviors
3rd	50 min	▪ Check the patient’s overall glycemic control status. ▪ Self-care record check (insulin, diet, exercise, special event, etc.). ▪ Check major glucose patterns. ▪ Check glucose patterns outside the target range. ▪ Education on problem factors according to diagnosed patterns. ▪ Diet and glucose control, exercise and glucose control, insulin adjustment. ▪ Set a glucose control target range (70–180 mg/dL). ▪ Set goals that you can achieve. ▪ Set a plan to utilize social support.	AGP report & leaflet	Health-promoting behaviors
5th	20 min	▪ Monitor changes such as whether goals are achieved and whether you are improving problematic aspects of your lifestyle and utilizing social support. ▪ If goals are not achieved or there are newly identified problems, additional interventions are implemented.	Phone call	Health-promoting behaviors
7th	20 min			
9th	20 min			
1th	20 min			
12th	20 min	▪ Check diabetes symptoms, social support, health beliefs, self-care, and physiological indices.	Questionnaire	Adaptive responses

Note. CGM = continuous glucose monitoring, AGP = ambulatory glucose profile.

**Table 2 healthcare-12-01381-t002:** Homogeneity test of participants’ general characteristics and variables (*N* = 50).

Characteristics	Categories	Exp. (*n* = 25)	Cont. (*n* = 25)	*t* or χ²	*p*
		*N* (%) or	*N* (%) or		
		M ± SD	M ± SD		
Sociodemographic characteristics					
Age (year)	<60	16 (64.0)	16 (64.0)	0.00	1.000
	≥60	9 (36.0)	9 (36.0)		
		53.96 ± 9.77	54.92 ± 11.73	−0.31	0.755
Sex	Male	11 (44.0)	10 (40.0)	0.08	0.500
	Female	14 (56.0)	15 (60.0)		
Education *	≤Middle school	9 (36.0)	4 (16.0)	2.60	0.196
	≥High school	16 (64.0)	21 (84.0)		
Occupation	Unemployed	9 (36.0)	8 (32.0)	0.09	0.500
	Employed	16 (64.0)	17 (68.0)		
Economic status	Low	5 (20.0)	6 (24.0)	0.12	0.500
	≥Middle	20 (80.0)	19 (76.0)		
Caregivers *	No	4 (16.0)	10 (40.0)	3.57	0.114
	Yes	21 (84.0)	15 (60.0)		
Drinking	No	20 (80.0)	19 (76.0)	0.12	0.500
	Yes	5 (20.0)	6 (24.0)		
Smoking	No	18 (72.0)	17 (68.0)	0.10	0.500
	Yes	7 (28.0)	8 (32.0)		
Disease-related characteristics					
Duration of disease (year)	<10	11 (44.0)	16 (64.0)	2.01	0.128
	≥10	14 (56.0)	9 (36.0)		
		12.39 ± 8.30	10.31 ± 8.12	0.90	0.372
Treatment modality	Insulin	9 (36)	10 (40.0)	0.09	0.500
	OHA + insulin	16 (64.0)	15 (60.0)		
Complication of DM	No	10 (40.0)	15 (60.0)	2.00	0.129
	Yes	15 (60.0)	10 (40.0)		
Comorbidity	No	7 (28.0)	6 (24.0)	0.10	0.500
	Yes	18 (72.0)	19 (76.0)		
Family history	No	9 (36.0)	10 (40.0)	0.09	0.500
	Yes	16 (64.0)	15 (60.0)		
Experience of hospitalization with DM	No	6 (24.0)	6 (24.0)	0.00	0.629
	Yes	19 (76.0)	19 (76.0)		
Experience of DM education (frequency)	1	13 (52.0)	19 (76.0)	3.13	0.070
	≥2	12 (48.0)	6 (24.0)		
		1.72 ± 0.89	1.32 ± 0.63	0.03	0.073
Main variables					
Previous health-promoting behavior		13.64 ± 3.87	14.84 ± 2.56	−1.29	0.203
Illness perception					
Consequences		7.72 ± 2.95	7.16 ± 2.29	0.75	0.457
Timeline		8.64 ± 2.56	8.92 ± 1.71	−0.46	0.651
Personal control		4.88 ± 2.37	5.20 ± 2.53	−0.46	0.647
Treatment control		6.92 ± 1.71	6.84 ± 2.48	0.13	0.895
Identity		6.16 ± 3.25	5.32 ± 2.39	1.04	0.304
Illness concern		6.92 ± 3.57	8.12 ± 1.99	−1.47	0.150
Coherence		5.44 ± 1.90	6.48 ± 1.87	−1.95	0.057
Emotional representation		4.00 ± 2.86	5.44 ± 3.19	−1.68	0.099
Diabetes symptoms					
Psychological–cognitive		16.80 ± 5.50	16.04 ± 6.75	0.44	0.664
Thirst–fatigue		12.56 ± 5.93	11.00 ± 4.06	1.09	0.284
Neurological		7.64 ± 3.07	7.80 ± 3.33	−0.18	0.860
Gastrointestinal comfort		11.04 ± 3.54	9.96 ± 3.80	1.04	0.304
Sexual		3.68 ± 2.54	2.68 ± 1.18	1.78	0.084
Diabetes social support		15.36 ± 7.61	18.92 ± 9.22	−1.49	0.161
Diabetes health beliefs					
Perceived susceptibility and severity		30.04 ± 6.49	33.04 ± 6.03	−1.69	0.097
Perceived barrier		11.08 ± 3.46	12.08 ± 3.17	−1.06	0.293
Perceived benefit		16.60 ± 2.94	15.92 ± 3.20	0.78	0.438
Self-care		56.36 ± 16.48	63.95 ± 11.30	−1.90	0.065
Maintenance		65.75 ± 15.23	69.83 ± 12.13	−1.05	0.300
Monitoring		50.59 ± 22.16	59.53 ± 16.49	−1.62	0.112
Management		34.67 ± 23.83	44.33 ± 14.06	−1.75	0.087
Confidence		74.45 ± 21.47	82.09 ± 18.00	−1.36	0.179
Physiological indices					
BMI (kg/m^2^)		24.74 ± 4.47	25.80 ± 3.75	−0.90	0.371
HbA1c (%)		10.76 ± 2.78	9.60 ± 1.92	1.71	0.093
FBS (mg/dL)		213.69 ± 80.55	186.76 ± 67.60	1.30	0.207
Cholesterol (mg/dL)		153.48 ± 35.15	169.76 ± 40.98	−1.51	0.138
LDL-C (mg/dL)		77.08 ± 28.01	93.44 ± 30.36	−1.98	0.053
HDL-C (mg/dL)		52.20 ± 13.76	53.72 ± 11.27	−0.43	0.671
TG (mg/dL)		142.84 ± 83.26	171.36 ± 119.56	−0.98	0.333
SBP (mmHg)		127.00 ± 14.73	129.20 ± 17.09	−0.49	0.628
DBP (mmHg)		74.12 ± 6.93	74.20 ± 11.32	−0.03	0.976

Note. Exp. = experimental group, Cont. = control group, M = mean, SD = standard deviation, DM = diabetes mellitus, OHA = oral hypoglycemic agents, BMI = body mass index, HbA1c = glycosylated hemoglobin, FBS = fasting blood sugar, LDL-C = low-density lipoprotein cholesterol, HDL-C = high-density lipoprotein cholesterol, TG = triglyceride, SBP = systolic blood pressure, DBP = diastolic blood pressure. * Fisher’s exact test.

**Table 3 healthcare-12-01381-t003:** Comparison of diabetes symptoms, diabetes social support, diabetes health beliefs, self-care, and physiological indices between groups (*N* = 50).

Variables		Group	Pre-Test	Post-Test	Difference	*t*	*p*
			M ± SD	M ± SD	M ± SD		
Diabetes symptoms	Psychological–cognitive	Exp. (*n* = 25)	16.80 ± 5.50	14.00 ± 4.93	−2.80 ± 2.58	−3.74	<0.001
		Cont. (*n* = 25)	16.04 ± 6.75	15.72 ± 6.13	−0.32 ± 2.08		
	Thirst–fatigue	Exp. (*n* = 25)	12.56 ± 5.93	10.00 ± 3.79	−2.56 ± 4.33	−2.03	0.049
		Cont. (*n* = 25)	11.00 ± 4.06	10.48 ± 3.72	−0.52 ± 2.57		
	Neurological	Exp. (*n* = 25)	7.64 ± 3.07	6.76 ± 3.02	−0.88 ± 1.76	−2.54	0.015
		Cont. (*n* = 25)	7.80 ± 3.33	7.67 ± 3.10	−0.17 ± 1.05		
	Gastrointestinal C omfort	Exp. (*n* = 25)	11.04 ± 3.54	8.76 ± 2.95	−2.28 ± 2.01	−3.75	<0.001
		Cont. (*n* = 25)	9.96 ± 3.80	9.76 ± 3.95	−0.20 ± 1.91		
	Sexual	Exp.(*n* = 25)	3.68 ± 2.54	3.68 ± 2.34	0.00 ± 1.80	0.00	1.000
		Cont. (*n* = 25)	2.68 ± 1.18	2.68 ± 1.22	0.00 ± 0.29		
Diabetes social support		Exp. (*n* = 25)	15.36 ± 7.61	18.12 ± 7.88	2.76 ± 3.96	2.95	0.005
		Cont. (*n* = 25)	18.92 ± 9.22	19.20 ± 9.45	0.28 ± 1.40		
Diabetes health beliefs	Perceived susceptibility and severity	Exp. (*n* = 25)	30.04 ± 6.49	31.44 ± 5.03	1.40 ± 3.55	1.46	0.152
		Cont. (*n* = 25)	33.04 ± 6.03	33.28 ± 6.54	0.24 ± 1.76		
	Perceived barrier	Exp. (*n* = 25)	11.08 ± 3.46	11.48 ± 3.38	0.40 ± 1.55	1.54	0.134
		Cont. (*n* = 25)	12.08 ± 3.17	11.96 ± 3.25	−0.12 ± 0.67		
	Perceived benefit	Exp. (*n* = 25)	16.60 ± 2.94	19.88 ± 2.55	3.28 ± 3.42	3.72	<0.001
		Cont. (*n* = 25)	15.92 ± 3.20	16.32 ± 2.84	0.40 ± 1.80		
Self-care	Maintenance	Exp. (*n* = 25)	65.75 ± 15.23	82.83 ± 13.82	17.08 ± 12.34	5.29	<0.001
		Cont. (*n* = 25)	69.83 ± 12.13	72.33 ± 12.21	2.50 ± 6.16		
	Monitoring	Exp. (*n* = 25)	50.59 ± 22.16	79.29 ± 13.60	28.71 ± 19.99	5.04	<0.001
		Cont. (*n* = 25)	59.53 ± 16.49	62.47 ± 18.43	2.94 ± 15.91		
	Management	Exp. (*n* = 25)	34.67 ± 23.83	62.67 ± 16.71	28.00 ± 21.52	4.42	<0.001
		Cont. (*n* = 25)	44.33 ± 14.06	49.89 ± 17.99	5.56 ± 13.51		
	Confidence	Exp. (*n* = 25)	74.45 ± 21.47	89.27 ± 13.94	14.82 ± 19.73	2.99	0.004
		Cont. (*n* = 25)	82.09 ± 18.00	84.00 ± 17.82	1.91 ± 8.79		
BMI (kg/m^2^)		Exp. (*n* = 25)	24.74 ± 4.47	25.48 ± 4.52	0.73 ± 1.92	−0.51	0.611
		Cont. (*n* = 25)	25.80 ± 3.75	27.15 ± 7.08	1.35 ± 5.73		
HbA1c (%)		Exp. (*n* = 25)	10.76 ± 2.78	7.86 ± 0.69	−2.89 ± 2.59	−3.83	<0.001
		Cont. (*n* = 25)	9.60 ± 1.92	9.19 ± 2.12	−0.41 ± 1.94		
FBS (mg/dL)		Exp. (*n* = 25)	213.68 ± 80.55	133.48 ± 35.26	−80.20 ± 79.56	−2.14	0.038
		Cont. (*n* = 25)	186.76 ± 67.60	151.48 ± 56.70	−35.28 ± 68.73		
Cholesterol (mg/dL)		Exp. (*n* = 25)	153.48 ± 35.15	152.60 ± 33.79	−0.88 ± 24.22	0.27	0.785
		Cont. (*n* = 25)	169.76 ± 40.98	166.64 ± 41.32	−3.12 ± 32.96		
LDL-C (mg/dL)		Exp. (*n* = 25)	77.08 ± 28.01	74.60 ± 26.57	−2.48 ± 16.98	0.49	0.624
		Cont. (*n* = 25)	93.44 ± 30.36	88.00 ± 30.58	−5.44 ± 24.70		
HDL-C (mg/dL)		Exp. (*n* = 25)	52.20 ± 13.77	55.36 ± 13.10	3.16 ± 9.39	2.39	0.021
		Cont. (*n* = 25)	53.72 ± 11.27	49.81 ± 14.72	−3.91 ± 11.39		
TG (mg/dL)		Exp. (*n* = 25)	142.84 ± 83.26	138.16 ± 86.72	−4.68 ± 53.99	0.45	0.654
		Cont. (*n* = 25)	171.36 ± 119.56	157.36 ± 117.48	−14.00 ± 88.12		
SBP (mmHg)		Exp. (*n* = 25)	127.00 ± 14.73	127.76 ± 13.18	0.76 ± 17.08	0.16	0.874
		Cont. (*n* = 25)	129.20 ± 17.09	129.24 ± 17.16	0.04 ± 14.88		
DBP (mmHg)		Exp. (*n* = 25)	74.12 ± 6.93	71.64 ± 9.30	−2.48 ± 10.27	0.00	1.000
		Cont. (*n* = 25)	74.20 ± 11.32	71.72 ± 9.39	−2.48 ± 11.55		

Note. Exp. = experimental group, Cont. = control group, M = mean, SD = standard deviation, BMI = body mass index, HbA1c = glycosylated hemoglobin, FBS = fasting blood sugar, LDL-C = low-density lipoprotein cholesterol, HDL-C = high-density lipoprotein cholesterol, TG = triglyceride, SBP = systolic blood pressure, DBP = diastolic blood pressure.

## Data Availability

All data generated or analyzed during this study are available on request from the corresponding author. The data are not publicly available due to domestic law.

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
