# Peer review of "Development and Effectiveness of a Pattern Management Educational Program Using Continuous Glucose Monitoring for Type 2 Diabetic Patients in Korea: A Quasi-Experimental Study"

_healthcare, 2024, doi:10.3390/healthcare12141381_

Round 1

Reviewer 1 Report

Comments and Suggestions for Authors
  • Introduction: here, the authors address the problem of glycemic uncontrol and its possible solutions through continuous glucose monitoring, now under the premise that this requires an educational intervention based on theory, which has not been previously studied. I consider that this section meets the criteria of precise definitions, magnitude, significance, previous studies and discrepancies that motivate the conduct of this study. I have no comments, observations, suggestions or recommendations to make.
  • Methods: The study design, selection criteria, sampling and sample size calculation are appropriate and well described. Also, the collection of sociodemographic, anthropometric and laboratory data is adequately explained. Also, the collection of sociodemographic, anthropometric and laboratory data is adequately explained. The description of the educational program, along with its theoretical and conceptual framework, are adequately explained, as well as the intervention and monitoring. The outcome variables are diabetes symptoms, social support, beliefs, self-care, and metabolic parameters. How the accuracy and reliability of the continuous glucose monitoring measurements were verified, as well as the description of the monitoring equipment (brand, model, country), are not mentioned.
  • Results: the characteristics of both groups were similar at the beginning of the study (Table 2 explains it). Table 2: If the proportions of age groups (< 60 and 60) are exactly equal, the p-value should be 1.000 (at this point, a two-tailed hypothesis is more adequate).
    Table 3: There seems to be a writing error in the "neurological symptoms" line of the control group: a negative sign is missing in the mean difference (it should say -0.17). The rest of variables showed a significant improvement at the post intervention time.
  • Discussion: The author compares the results of his study related to perceived social support with those of another study (Azami), and a meta-analysis (Song). Likewise, he highlights the main findings in the improvement of other self-care and physiological parameters. The limitations are sufficiently stated, and conclusions are derived from the results.

Author Response

We appreciate the time and effort you and the reviewers have put into providing valuable feedback and insightful comments, improving our manuscript. We have carefully considered each comment and updated the manuscript, as required. We have marked the revisions made to the manuscript in red font.

Reviewer 2 Report

Comments and Suggestions for Authors

Since no previous studies applied and verified the theory that was developed by Whittemore and Roy two decades ago, Seung-Yeon Kong et al aimed at developing a program based on the adaptation to the diabetes mellitus middle-range theory of Whittemore and Roy for patients with diabetes. Their current study aimed at developing a pattern management educational program using CGM to improve self-care, based on Whittemore and Roy’s theory, in patients with diabetes and to verify its effectiveness. It has been almost more than 2 decades since the theory was developed and what interest you to do in this area? I mean it is quit long period. How you develop and verify/ validate in one study particularly using the same population (sample). Even your sample size is very small. What are your inclusion and exclusion criteria? Please put a clear and precise inclusion and exclusion criteria. The quality of figures is very poor, difficult to see them. Table 1 is poor quality, for example the weeks of ‘’step 4’’ is not clear, difficult to read. 

Author Response

(The authors gave the same response as above.)

Round 2

Reviewer 2 Report

Comments and Suggestions for Authors

Thank you very much for your response in addressing the points raised in my review. Your response to ‘’point 2’’ is not convincing. You are correct when it comes to estimating sample size. However, you did not say anything about the point I raised- how do you validate/ verify in the same/ one sample or population?  You test the hypothesis then you validate it on another sample or population. I am not convinced about this point. It will mislead the readers.

Still, table 3 (in the response to the reviewer's cover letter) is blurred, and difficult to read. besides, the quality of Table 1 and Figure 2 is low. please check it again. 

In the iThenticate report, the amount of wording duplication in the manuscript is a bit high. Could you please decrease the amount of wording duplicates?

Author Response

We appreciate the time and effort you and the reviewers have put into providing valuable feedback and insightful comments, improving our manuscript. We have carefully considered each comment and updated the manuscript, as required. We have marked the revisions made to the manuscript in red font.

We attach the response as a file.
